# Factors Associated with ICU Admission in Patients with COVID-19: The GOL2DS Score

**DOI:** 10.3390/medicina57121356

**Published:** 2021-12-12

**Authors:** Marcello Candelli, Giulia Pignataro, Miriana Ferrigno, Sara Cicchinelli, Enrico Torelli, Antonio Gullì, Marta Sacco Fernandez, Andrea Piccioni, Veronica Ojetti, Marcello Covino, Antonio Gasbarrini, Massimo Antonelli, Francesco Franceschi

**Affiliations:** 1Emergency Medicine Department—Fondazione, Universitaria Policlinico Agostino Gemelli–IRCCS–Catholic, University of Sacred Heart of Rome, 100168 Rome, Italy; giulia.pignataro@policlinicogemelli.it (G.P.); ferrignomiriana@gmail.com (M.F.); sara.cicchinelli@policlinicogemelli.it (S.C.); enrico.torelli@policlinicogemelli.it (E.T.); martasaccofernandez@gmail.com (M.S.F.); andrea.piccioni@policlinicogemelli.it (A.P.); veronica.ojetti@policlinicogemelli.it (V.O.); marcello.covino@policlinicogemelli.it (M.C.); francesco.franceschi@policlinicogemelli.it (F.F.); 2Emergency, Anesthesiological and Reanimation Sciences—Fondazione, Universitaria Policlinico Agostino Gemelli–IRCCS–Catholic, Faculty of Medicine, University of Sacred Heart of Rome, 100168 Rome, Italy; antonio.gulli@policlinicogemelli.it (A.G.); massimo.antonelli@policlinicogemelli.it (M.A.); 3Department of Medical and Surgical Sciences—Fondazione, Universitaria Policlinico Agostino Gemelli–IRCCS–Catholic, University of Sacred Heart of Rome, 100168 Rome, Italy; antonio.gasbarrini@policlinciogemelli.it

**Keywords:** COVID-19, prognostic score, ICU

## Abstract

*Background and Objectives*: The COVID-19 pandemic has been shaking lives around the world for nearly two years. The discovery of highly effective vaccines has not been able to stop the transmission of the virus. SARS-CoV-2 shows completely different clinical manifestations. A large percentage (about 40%) of admitted patients require treatment in an intensive care unit (ICU). This study investigates the factors associated with admission of COVID-19 patients to the ICU and whether it is possible to obtain a score that can help the emergency physician to select the hospital ward. *Materials and Methods*: We retrospectively recorded 313 consecutive patients who were presented to the emergency department (ED) of our hospital and had a diagnosis of COVID-19 confirmed by polymerase chain reaction (PCR) on an oropharyngeal swab. We used multiple logistic regression to evaluate demographic, clinical, and laboratory data statistically associated with ICU admission. These variables were used to create a prognostic score for ICU admission. Sensitivity, specificity, positive predictive value (PPV), negative predictive value (NPV), and receiver-operating characteristic curve (ROC) of the score for predicting ICU admission during hospitalization were calculated. *Results*: Of the variables evaluated, only blood type A (*p* = 0.003), PaO2/FiO2 (*p* = 0.002), LDH (*p* = 0.004), lactate (*p* = 0.03), dyspnea (*p* = 0.03) and SpO2 (*p* = 0.0228) were significantly associated with ICU admission after adjusting for sex, age and comorbidity using multiple logistic regression analysis. We used these variables to create a prognostic score called GOL2DS (group A, PaO2/FiO2, LDH, lactate and dyspnea, and SpO2), which had high accuracy in predicting ICU admission (AUROC 0.830 [95% CI, 0.791–0.892). *Conclusions*: In our single-center experience, the GOL2DS score could be useful in identifying patients at high risk for ICU admission.

## 1. Introduction

In late 2019, a new coronavirus responsible for a cluster of pneumonia cases in Wuhan, in China’s Hubei region, was identified. The pneumonia was named Coronavirus Disease 2019 (COVID-19) and the virus was named Severe Acute Respiratory Syndrome Coronavirus 2 (SARS-CoV-2) [1]. It spread rapidly throughout the world and has caused more than 250,000,000 infections and about 5,000,000 deaths to date. SARS-CoV-2 is a single-stranded positive RNA virus that spreads among humans through respiratory droplet infection [2]. The virus can spread not only through respiratory secretions but also through other biological pathways such as the oral-fecal route [3,4]. The diagnostic standard is identification of the virus in a nasopharyngeal swab by real-time polymerase chain reaction test (RT-PCR). COVID-19 can present in a variety of shapes, from asymptomatic to fatal ones [4]. Typical clinical symptoms include fever, dry cough, dyspnea, headache and pneumonia. The latter is related to damage to the alveoli, which can lead to progressive respiratory failure. In the February 2020 Chinese CDC report (*n*=) [5], three levels of severity were distinguished. Mild disease (81%) shows no or mild pneumonia. Severe disease (14%) is characterized by dyspnea, respiratory rate ≥ 30/min, peripheral blood oxygen saturation SpO2 ≤ 93%, PaO2/FiO2 < 300 and/or lung infiltration > 50%. Critical illness (5%) presents with respiratory failure, septic shock and/or multiple organ failure (MOD) or multiple organ failure (MOF). Pre-existing conditions, clinical and laboratory features may predict disease progression. For example, older age, a D-dimer concentration greater than 1 µg/mL, and a high SOFA-score at the time of hospitalization are associated with a higher risk of in-hospital death [6]. In addition, higher levels of IL-6, high-sensitivity troponin-I and lactate dehydrogenase are more frequently associated with more severe COVID-19 [7,8,9]. Almost 90% of patients with pneumonia had hypercoagulability with higher D-dimer concentrations. Higher D-dimer concentrations are associated with higher mortality [9]. Deep vein thrombosis (DVT) has been described in COVID-19 patients, especially in the ICU, even despite previous performance of thromboprophylaxis [10,11,12]. Earlier vaccination has also been associated with better disease outcome [13]. Several factors have been associated with severity of illness, but a prognostic score may be useful for emergency physicians to properly evaluate admission to the ward. CURB-65 is a score which is widely used in emergency medicine to stratify the risk of death in patients with community-acquired pneumonia. It assesses five simple variables: presence of confusion, blood urea nitrogen, respiratory rate, blood pressure, and age. Since COVID-19 typically presents as interstitial pneumonia, CURB-65 was also used to assess the severity of COVID-19. However, the results were inconsistent. The aim of this study is to develop a clinical prognostic score to predict the need for intensive care unit (ICU) admission in patients admitted for COVID-19 and to compare it with CURB-65.

## 2. Materials and Methods

This is a monocentric, observational, retrospective study. We included all patients older than 17 years who were hospitalized by the Emergency Department of Fondazione Policlinico Gemelli. IRCCS, Catholic University of the Sacred Heart of Rome, among those who were admitted with SARS-CoV-2 infection in March 2020. The diagnosis of SARS-CoV-2 was confirmed with RT-PCR on nasopharyngeal and/or oropharyngeal swabs. The study was approved by our Academic Ethics Committee (protocol number 0023001/20) and consent to participate in the study and to process personal data was obtained in accordance with the provisions of Italian laws and Italian data protection authorities. Among the initially included patients, we excluded patients with mixed infections (i.e., COVID-19 and other infectious diseases), patients with COVID-19 and other prevalent acute conditions affecting prognosis (i.e., surgical emergencies, time-dependent diseases, pathologies with high risk of fatal outcome) and patients discharged from or who died at emergency room (ER) We divided patients into two groups:Hospitalized patients who did not require admission to ICU during their hospital stay (medical ward patients–MWP);Hospitalized patients requiring admission to ICU during their hospital stay (ICU patients–ICUP).

For all patients, we collected various kinds of data in the ED:Demographic: age, gender, chronic diseases, blood type;Clinical manifestations: presentation signs and symptoms, vital signs;Laboratory: arterial blood gas analysis, complete blood count, electrolytes, renal and liver function tests, C-reactive protein and procalcitonin, high sensitivity troponin, N-terminal pro-bone natriuretic peptide (Nt-pro-BNP);CURB-65, a prognostic score for patients with community acquired pneumonia, was calculated for all enrolled patients

We used the software IBM SPSS STATISTICS 20 for statistical analysis. Data were analyzed using descriptive statistics. Categorical values were expressed as % of total, other values as mean ± standard deviation (SD) or with median and range, the latter of which only in case of parametric data. Continuous variables were compared using the Student *t*-test. Categorical variables were compared using the chi-square test or Fisher’s exact test when appropriate. All variables with a *p* value of less than 0.2 were included in the multivariate logistic regression analysis after adjustment for confounding factors (age, sex, comorbidities). A value of *p* < 0.05 was considered statistically significant. For each factor that remained significantly associated with ICU admission after multivariate analysis, an analysis of the receiving operative characteristic (ROC) was also performed. Using the cutoff values with higher sensitivity or specificity for each variable, we created a prognostic score. Then, we calculated the sensitivity, specificity, positive predictive value (PPV), negative predictive value (NPP) and best cutoff value by creating a ROC curve of the prognostic score. Finally, we compared the accuracy of the prognostic score with CURB65.

## 3. Results

We enrolled 313 patients, of which 210 were male and 103 were female. The mean age was 64.9 ± 15 years. We observed a 30-day mortality rate of 16.6% (52/313). Regarding smoking habits, 57% were non-smokers, 7% were smokers, and 36% were ex-smokers. Patients differed in the presence or absence of comorbidities and in the number of comorbidities. 32.9% (103) of patients had no comorbidities (Table 1), 29.1% (91) had one comorbidity, 15.3% (48) had two, 10.8% (34) had three, 8.0% (25) had four and 3.8% (12) had five or more comorbidities. The most common comorbidities were systemic arterial hypertension (46.7%), ischemic cardiomyopathy (13.9%), and diabetes mellitus (12.7%). Other common diseases were obesity, COPD, active neoplasms, chronic heart failure, atrial fibrillation, valvular heart disease, Parkinson’s disease, and Alzheimer’s disease. Among chronic diseases, we need to mention chronic thyroid diseases such as thyroiditis, hypothyroidism and hyperthyroidism, hypercholesterolemia, dyslipidemia, chronic renal failure, benign prostatic hyperplasia, depression, dementia, epilepsy, ulcerative recto-colitis, and thrombocytopenia. Among neoplasms, colorectal cancer, breast cancer, lung cancer, and leukemia were the most common.

The most common symptoms were fever (*n* = 302; 96.1%), dry cough (*n* = 225; 71.6%), dyspnea (*n* = 223; 71.0%), asthenia (*n* = 176; 56.0%), anorexia (*n* = 110; 35.0%), dysgeusia/ageusia (*n* = 93; 29.6%), dysosmia/anosmia (*n* = 86; 27.43%), and arthralgia/arthritis (*n* = 70; 22.3%). Less common manifestations, in order of frequency, were myalgia (22.3%), nausea (16.9%), headache (16.0%), conjunctivitis (12.6%) and pharyngitis (12.6%). Many patients required supplemental oxygen. Most patients (N: 143, 45.5%) required a nasal cannula, venturi mask (VM) or non-rebreathing face mask (NRB) with a FiO2 between 0.24 and 1 (median FiO2 0.24). Eighteen (5.7%) patients required oxygen therapy with high-flow nasal cannulae (HFNC). Ten (3.2%) patients were ventilated with non-invasive ventilation (NIV) using a helmet and 16 (5.1%) patients were ventilated with a full-face mask. Orotracheal intubation (OTI) was required in 44 (14%) patients.

77% (232) of patients admitted to a medical ward did not require intensive care. 25.8% (81) required at least 1 day of ICU stay. The mean hospital length of stay (hLOS) in the medical ward was 16.1 ± 11.3 days whereas in the ICU it was 27.3 ± 20.6 days.

### 3.1. Comparison of Groups

#### Univariate Analysis of MWP versus ICUP

Most of the demographic characteristics did not reach statistical significance, except for blood group A and age (Table 1). No statistically significant difference was found when comparing symptoms, except for nausea, which was reported in 18.5% of MWP and in 4% of ICUP (*p* = 0.02) and dyspnea, which was reported in 65% of MWP and 76.5% of ICUP (*p* = 0.004). Only a few comorbidities were found to have a statistically significant different prevalence between the two groups. Systemic arterial hypertension was observed in 43% of MWP and 58% of ICUP (*p* = 0.02). Ischemic cardiomyopathy was noted in 10% and 25% of MWP and ICUP, respectively (*p* = 0.0007). Obesity was observed in 10% of MWP and 19% of ICUP (*p* = 0.03). As for vital signs, arterial blood gas analysis and laboratory findings, the parameters that reached different degree of statistically significant association with ICU admission and can be found in Table 2.

### 3.2. Multivariate Analysis

Multiple logistic regression was used to perform a multivariate analysis to evaluate the statistically significant differences between the MWP and ICUP, applying corrections for different confounding factors (age, gender and comorbidities). We included all variables in the model with a *p* < 0.2 in the univariate analysis. The variables that resulted statistically and independently significant in the multivariate analysis were blood group A, PaO2/FiO2, LDH, lactate, dyspnea, and SpO2 (Table 3).

A ROC curve analysis was performed and we calculated the AUC (Figure 1, Figure 2, Figure 3 and Figure 4) for all non-dichotomous variables. The AUC was at least 0.699, indicating that these parameters have good accuracy in predicting ICU admission (Figure 1, Figure 2, Figure 3 and Figure 4). Afterwards, we assigned a score to each variable with cut-off values with high sensitivity or specificity (Table 4).

These individual scores were combined into a single score system called GOL2DS (blood type A, oxygenation, lactate, LDH, dyspnea, SpO2). ROC was also calculated for the GOL2DS scoring system. The accuracy of GOL2DS was higher than that of the scores calculated for each variable (Figure 5).

We also calculated the CURB-65 score and performed ROC curve analysis to evaluate the accuracy of CURB-65 in predicting ICU admission (Figure 6). Accuracy was acceptable, although with a lower value than the GOLD2S score (0.678). GOL2DS score has high sensitivity for the highest scores and high specificity for lower scores. Moreover, it has a high positive predictive value, reaching 100% for lower scores and a high negative predictive value for the highest scores (Table 5). We calculated GOL2DS score for all included patients to estimate the likelihood that a patient admitted to a medical ward would require admission to the ICU (Table 4). A GOL2DS score < 1 showed a very low risk of ICU admission with a high specificity (92.59). In contrast, a GOL2DS score > 3 showed high sensitivity (>95) and identified patients at very high risk of needing ICU admission.

## 4. Discussion

In our cohort of COVID-19 patients, men were shown to be more likely to be hospitalized than women (67% vs. 33%). Many hypotheses attempt to explain the sex difference in COVID-19 and we cannot rule out the possibility that the difference depends on confounding factors such as the number of performed nasopharyngeal swabs, which may be higher in men. In addition, women may have been less exposed to the virus than men during a period of “strong” lockdown, such as March 2020 in Italy. In Italy, a deeply rooted cultural and traditional factor for women to use parental leave during school closures allows them not to work to take care of children largely than men. On the other hand, there could be differences between men and women’s immune systems. These differences could be either genetic, such as genotypic differences in chromosomes XX and XY, or hormonal, controlled by estrogens and progesterone in women and by androgens in men [14]. A recent European study reported similar infection rates in males and females, but higher severity in males. [15]. Therefore, the higher severity of COVID-19 in men may partly explain the higher prevalence of males in our cohort, which only includes hospitalized patients.

The most common comorbidity in our study was systemic arterial hypertension. These data are consistent with medical literature, as is the observation of a low prevalence of other comorbidities such as diabetes mellitus, atrial fibrillation, chronic heart failure, COPD and others [16]. The clinical presentation of SARS-CoV-2 infection is similar to many other viral infections, with the exception of dysgeusia/dysosmia and conjunctivitis, which are highly suggestive of COVID-19. Laboratory findings are also similar to those of other viral infections, but with some notable features. The most frequently altered laboratory values in our cohort were blood urea nitrogen, which can be explained by dehydration and D-dimers, whose increase can be explained by the inflammation and thrombotic events that characterize COVID-19. The latter can reach extremely high levels, which are unusual in other viral diseases. In addition, relative lymphopenia, an increase in fibrinogen and Nt-proBNP levels were also observed. Chest radiographic findings are similar to other viral diseases, especially influenza virus. As mentioned above, COVID-19 can have heterogeneous clinical courses, ranging from asymptomatic forms to severe respiratory failure that can rapidly lead to death. In our cohort, we observed a 30-day mortality rate of 16.6% (52/313). We focused on hospitalized patients and examined the presence of statistically significant differences between clinical, demographic and laboratory variables. Univariate analysis showed that the ICUP were more likely to be male, obese, affected by hypertension and ischemic cardiomyopathy, with a blood group A and more likely to suffer from nausea and dyspnea. They also had lower SpO2 and PaO2/FiO2 ratio and higher levels of lactate, blood urea nitrogen, CPK, LDH, NT-proBNP, and total neutrophils. Other parameters such as blood pH, hemoglobin, procalcitonin and troponin showed a tendency to associate with the intensity of healing, but without reaching statistical significance. These data suggest that patients with a more severe clinical course have more lung involvement, suggested by both higher LDH levels [17] and worse arterial blood gas parameters and a higher prevalence of bacterial superinfection, suggested by higher levels of lactate, total neutrophils and procalcitonin. In addition, other organs also appear to be involved. For example, higher CPK levels suggest muscle involvement, whereas elevated levels of blood urea nitrogen, troponin and NT-proBNP and suggest renal and cardiac involvement. In our study, only 6 variables evaluated at presentation in the ED showed statistically significant association with ICU hospitalization: blood group A, alveolar oxygenation (PaO2/FiO2), lactate, LDH, dyspnea, SpO2. All these variables showed high values for sensitivity and specificity. The association with respiratory parameters was not surprising. The association with blood group A was curious, so we investigated the reasons behind this finding. The reasons seem to be multiple. First, an association between a more severe disease development and group A was found in two Chinese studies [18,19]. Blood group A is associated with higher expression of ACE2 receptors, while group 0 is less sensitive to infection due to the presence of anti-A antibodies, which may reduce the availability of ACE2 receptors for virus binding. Patients with low GOL2DS scores had a low probability of being hospitalized in the ICU. We can hypothesize that this group of patients (GOL2DS score > 1) can be directly admitted to a low intensity ward. Patients with higher GOL2DS scores (>3) had a high probability of being hospitalized in the ICU in the following days, so they should be closely monitored or admitted to a sub-intensive ward. Other scores have been proposed to assess COVID-19 severity. These include the CALL and the CHOSEN scores. [20,21]. The CALL score identifies patients at risk of disease progression based on comorbidity (without 1 point, with 4 points), age (≤60 years 1 point, >60.3 points), lymphocyte count (≤1.0 × 10^9^/L 1 point, >1 × 10^9^/L 3 points) and LDH (≤250 U/L 1 point, 250–500 U/L 2 points, >500 U/L 3 points). The accuracy of the score was 0.91 with an NPV of 98.5% considering a cut-off score < 7. However, the endpoint of the study was not ICU admission but disease progression defined as worsening respiratory rate (>30 breaths/min), oxygen saturation (<93%), PaO2/FiO2 (<300 mmHg) and pulmonary CT findings or need for mechanical ventilation during hospitalization. Moreover, other external studies failed to validate this score [22,23,24]. The Chosen score (COVID Home Safely Now) uses three parameters (age, SpO2 and albumin) and has been used along with an endpoint (ICU admission, oxygen requirement, or death at 14 days) to discharge patients safely [21]. It follows the opposite goals of our score, which was proposed to select patients who need close monitoring since they are at higher risk of deterioration. Moreover, the CHOSEN score has not been validated by an external study [22]. The ANDC score and the HA2T2 were proposed to predict mortality and have not been validated by external studies [21]. Another interesting study compared classical risk scores (pneumonia severity index, CURB, CURB-65, qSOFA, ReA-icu, SCAP) and new scores proposed to assess the severity of COVID-19 (COVID GRAM, CALL and 4C) to predict both mortality and ICU admission [23]. The authors found that the classic and new scores were comparable in terms of 30-day mortality. In contrast, none of the scores assessed achieved acceptable accuracy (AUC > 0.650) in predicting ICU admission. In our cohort, CURB-65 showed good accuracy in predicting ICU admission, but it was less accurate than GOL2DS. Our GOL2DS score is a promising tool for identifying patients requiring ICU admission, but external validation is needed.

## 5. Limitations of the Study

Our study has several limitations. First, it is a retrospective study with all the limitations associated with its design. In addition, we decided to exclude patients discharged from the ER and patients who died in the ER since we were looking for a score that could help the emergency physician properly choosing the intensity of ward and observation that patients need. The patients who were discharged from ER and died there did not need observation and admission. On the other hand, this could be interpreted as selection bias. Nevertheless, we believe that excluding the extremes improves the utility of the score. It should only be used in patients who require hospitalization (if validated by future studies). Another limitation is the lack of a validation group. The number of patients included did not allow us to split our population into two cohorts. However, we believe that external validations have a higher value than internal ones. Indeed, all scores proposed for ICU admission in COVID-19 patients failed external validation, even if they had high accuracy in the internal validation cohort [21,22,23,24]. Moreover, this is a monocentric study and our results cannot be generalized to all situations. It is important to emphasize that, unlike other Italian regions, intensive care beds were always available in our hospital for COVID-19 patients from our ER and other secondary and primary care hospitals, even during March 2020.

## 6. Conclusions

The GOL2DS score indicates the probability that the patients might require hospitalization in the ICU. It has high sensitivity and specificity and high positive predictive value. It could be a valid tool in the management of COVID-19 patients that deserve hospital admission. Further external studies are needed to validate this score and to generalize the results to other settings.

## Figures and Tables

**Figure 1 medicina-57-01356-f001:**
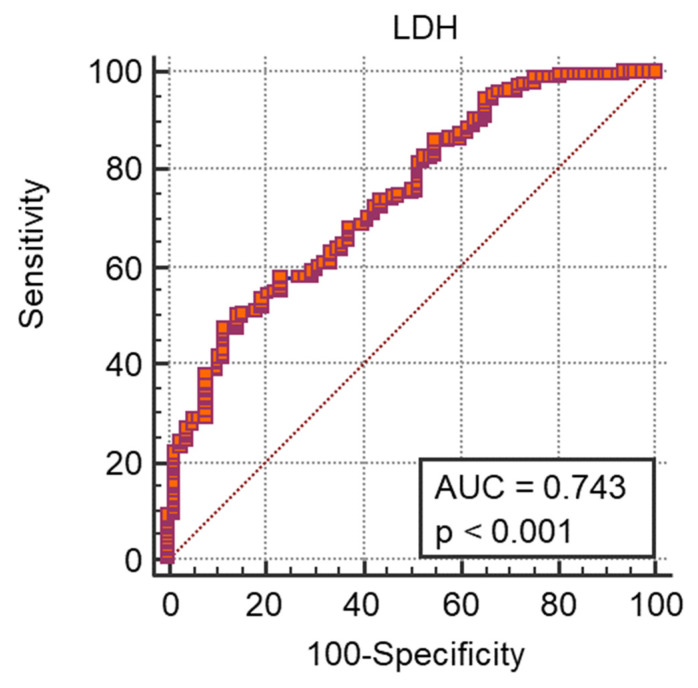
ROC curve e AUC values for LDH and ICU admission.

**Figure 2 medicina-57-01356-f002:**
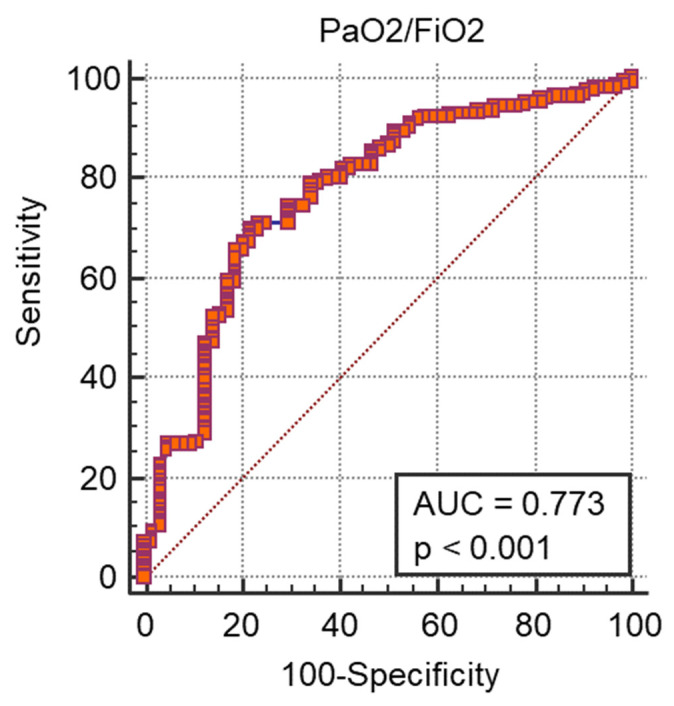
ROC curve e AUC values for PaO2/FiO2 and ICU admission.

**Figure 3 medicina-57-01356-f003:**
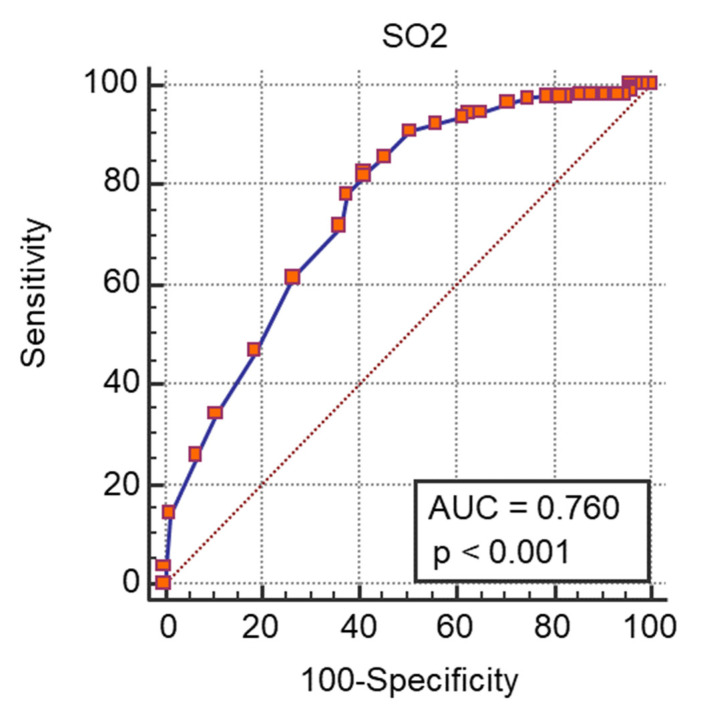
ROC curve e AUC values for SO_2_ and ICU admission.

**Figure 4 medicina-57-01356-f004:**
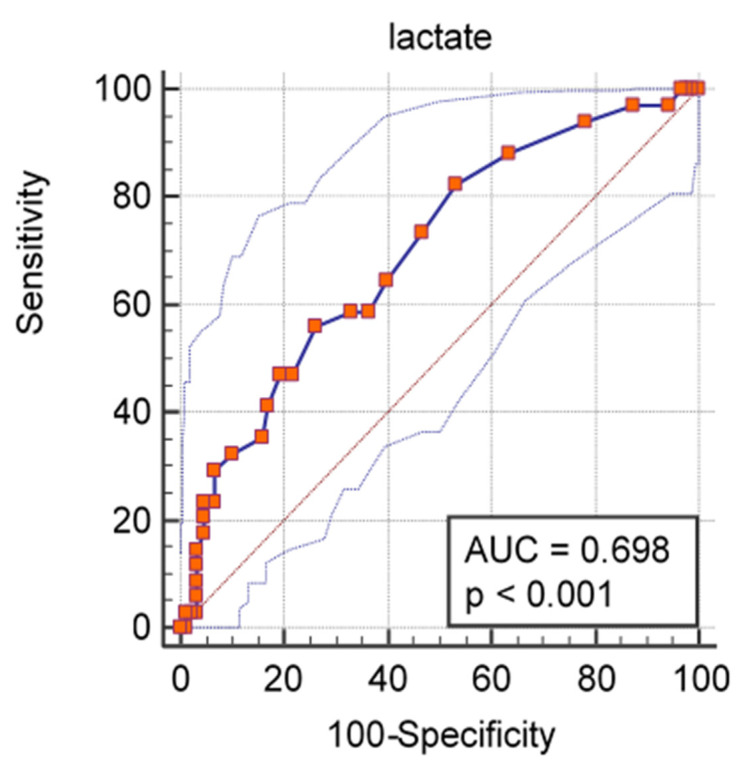
ROC curve and AUC values for lactate and ICU admission.

**Figure 5 medicina-57-01356-f005:**
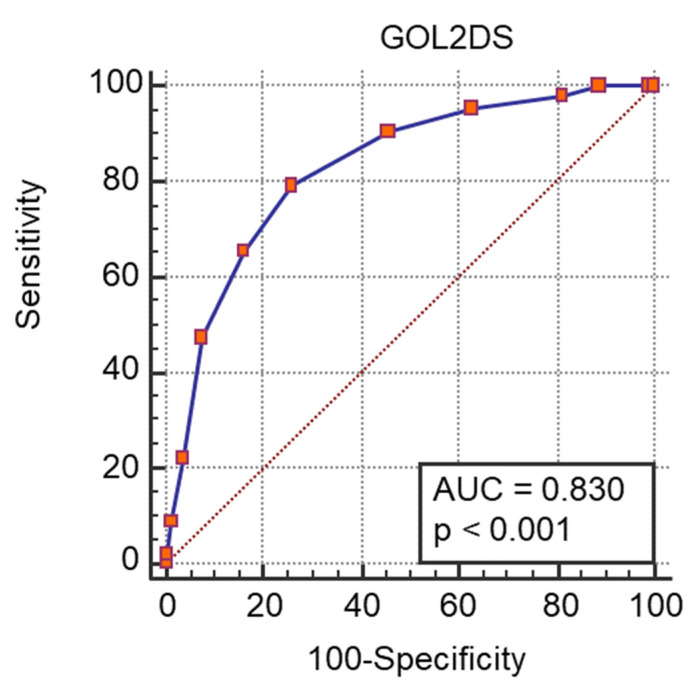
ROC curve and AUC values for GOL2DS score and ICU admission.

**Figure 6 medicina-57-01356-f006:**
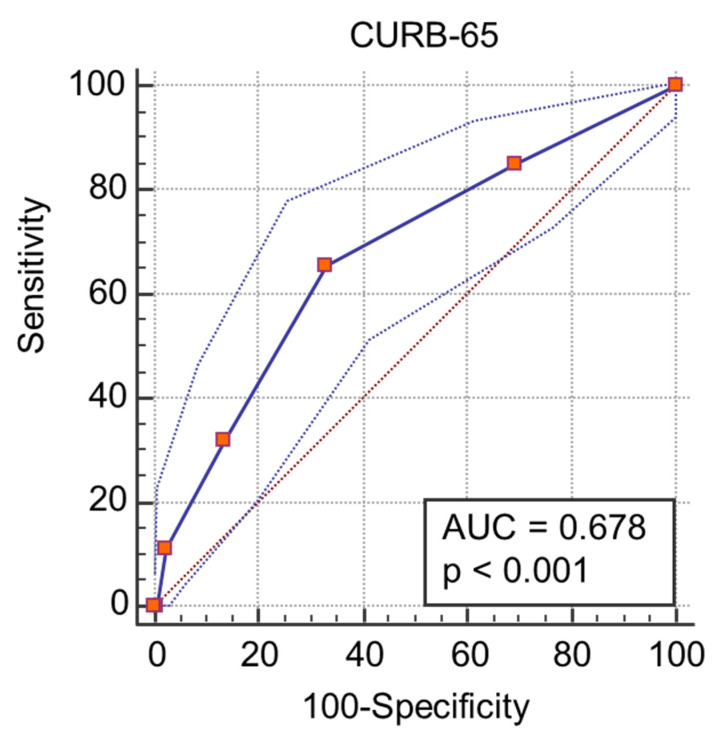
ROC curve and AUC values for CURB-65 score and ICU admission.

**Table 1 medicina-57-01356-t001:** Demographic and clinical data of enrolled patients and comparison between groups.

	All Patients	MWP	ICUP	*p*
Sex (Male)	210/313 (67.1)	152/232 (65.6)	58/81 (71.6)	0.31
Smokers	22/313 (7.0)	14/232 (6)	8/81 (9.8)	0.24
Age (mean ± SD)	65 ± 15	64 ± 16	69 ± 14	<0.01
No comorbidity	103/313 (32.9)	83/232 (35.7)	20 (24.7)	0.07
ARB/ACEi	111/313 (35.5)	78/232 (33.6)	33/71 (40.7)	<0.05
Blood Group A	109/311 (35.0)	77/232 (33.2)	32/69 (46.4)	<0.05
Rh (-) factor	70/311 (22.5)	56/242 (23.14)	14/69 (20.3)	0.51

Data are presented as Numbers/Total (%) or as mean ± SD. Age was calculated in years. MWP, medical ward patients; ICUP, intensive care unit patients; ARB, angiotensin receptors blockers; ACEi, angiotensin-converting enzyme inhibitors. Variables in bold have been included in multivariate analysis.

**Table 2 medicina-57-01356-t002:** Comparison of vital signs, arterial blood gas analysis, and the laboratory findings between groups.

	All Patients	MWP	ICUP	*p*
SpO2 (%)	92.7 ± 7	94.4 ± 5	87.8 ± 10	<0.001
PaO2 (mmHg)	72.4 ± 23	74.9 ± 24	63.5 ± 20	<0.01
PaCO2 (mmHg)	33.5 ± 6	33.8 ± 5	32.3 ± 9	0.39
Bicarbonate (Mmol/L)	23.7 ± 3	23.9 ± 3	23.5 ± 3	0.42
Lactate (mEq/L)	2.0 ± 4	1.7 ± 4	4 ± 6	<0.05
Chloride (mEq/L)	102 ± 10	102 ± 11	102 ± 6	0.78
PaO2/FiO2	298 ± 100	317 ± 94	232 ± 89	<0.0001
pH	7.4 ± 0.05	7.4 ± 0.05	7.4 ± 0.08	0.45
SBP (mmHg)	127 ± 26	130 ± 23	126 ± 24	0.79
DBP (mmHg)	77 ± 15	77 ± 14	76 ± 18	0.26
HR (BPM)	93 ± 19	95 ± 19	92 ± 21	0.21
RR (BPM)	27 ± 29	32 ± 2	23 ± 12	0.54
BUN (mg/dL)	22 ± 17	20 ± 16	28 ± 17	<0.003
Creatinine (mg/dL)	1.6 ± 6	1.1 ± 1	1.7 ± 7	0.31
Sodium (mEq/L)	138 ± 5	137 ± 6	138 ± 5	0.22
Potassium (mEq/L)	4.0 ± 0.5	4.2 ± 0.5	3.9 ± 0.5	0.41
ALT (UI/L)	39 ± 45	41 ± 29	38 ± 49	0.34
AST (UI/L)	98 ± 536	54 ± 47	113 ± 621	0.09
Bilirubin (mg/dL)	0.7 ± 0.4	0.7 ± 0.3	0.7 ± 0.5	0.43
CPK (UI/L)	216 ± 344	190 ± 281	300 ± 496	0.02
LDH (UI/L)	377 ± 416	348 ± 448	473 ± 261	0.03
CRP (mg/L)	95 ± 85	80 ± 74	143 ± 100	<0.0001
Procalcitonin (ng/mL)	0.9 ± 4	0.8 ± 4	1.1 ± 5	0.54
Hb (g/dL)	13.9 ± 2	13.9 ± 2	13.8 ± 2	0.97
Platelets (×10^9^/L)	200 ± 76	198 ± 86	201 ± 73	0.99
Neutrophils (×10^9^/L)	5.313 ± 3.142	4.975 ± 3.045	6.449 ± 3.215	<0.001
Lymphocytes (×10^9^/L)	1.266 ± 2.330	1.237 ± 1.467	1.362 ± 3.563	0.83
D-Dimer (ng/mL)	2780 ± 5400	2496 ± 5247	3550 ± 5772	0.21
Troponine (ng/L)	875 ± 5731	123 ± 628	3134 ± 9856	0.01
Nt-proBNP (pg/mL)	2025 ± 5261	1814 ± 5139	2651 ± 5655	0.55

Data are presented as Numbers/Total (%) or as mean ± SD. SAP, systolic blood pressure; DBP, diastolic blood pressure; HR, heart rate; RR, respiratory rate; BUN, blood urea nitrogen; ALT, alanine aminotransferase; AST, aspartate aminotransferase; CPK, creatine phosphokinase; LDH, lactate dehydrogenase; CRP, C reactive protein; Hb, hemoglobin; Nt-proBNP, N-terminal pro-brain natriuretic peptide. Variables which were included in multivariate analysis are in bold.

**Table 3 medicina-57-01356-t003:** Odds ratio and 95% confidence interval for variables significantly associated with ICU admission after multiple logistic regression and correction for sex, age, and comorbidities.

	Odds Ratio	[95% IC]	*p*
Blood Group A	2.937	1.642–5.111	<0.005
PaO2/FiO2	5.720	1.901–9.329	<0.001
Lactate	1.481	1.175–2.366	<0.05
LDH	3.017	2.391–3.958	<0.005
Dyspnea	3.269	1.889–5.441	<0.01
SpO2	2.006	1.547–2.636	<0.01

**Table 4 medicina-57-01356-t004:** GOL2DS score. Best cut off sensibility and specificity values for lactate, PaO2/FiO2, LDH and SpO2. A score was assigned for each cut off value. The sum of the scores for each variable determines the GOL2DS score.

Variable	Sensitivity	95% CI	Specificity	95% CI	+PV	−PV	Score
Lactate ≤ 0.7	21.6	13–31	94.6	80–99	39.3	87.2	−1
Lactate > 2	89.9	81–95	32.3	17–50	19.0	94.7	1
Group A							1
Other Group							−1
PaO2/FiO2 < 200	93.0	88–96	37.5	25–50	20.8	96.8	2
PaO2/FiO2 < 300	67.4	60–74	79.7	67–88	36.9	93.3	1
PaO2/FiO2 > 360	29.1	22–36	87.5	77–94	29.1	87.5	−1
Dyspnea							1
LDH ≤ 250	29.0	23–35	94.9	87–99	50.0	88.3	−1
LDH > 500	90.2	85–94	37.2	26–49	20.2	95.5	2
SpO2 < 90	85.7	80–90	54.7	42–66	25.0	95.6	2
SpO2 90–94	61.6	54–68	73.3	62–83	29.0	91.5	1
SpO2 > 98	14.3	10–20	98.7	93–100	65.4	86.7	−1

**Table 5 medicina-57-01356-t005:** Sensitivity, specificity, predictive positive, and negative values of GOL2DS score in predicting ICU admission.

Criterion	Sensitivity	95% CI	Specificity	95% CI	+PV	−PV
<−4	0	0.0–1.6	100	95.5–100	100	25.9
<−3	1.72	0.5–4.4	100	95.5–100	100	26.2
≤−2	8.62	5.3–13.0	98.77	93.3–100	95.2	27.4
≤−1	21.98	16.8–27.9	96.3	89.6–99.2	94.4	30.1
≤0	47.41	40.8–54.1	92.59	84.6–97.2	94.8	38.1
≤1	65.52	59.0–71.6	83.95	74.1–91.2	92.1	45.9
≤2	79.31	73.5–84.3	74.07	63.1–83.2	89.8	55.6
≤3	90.52	86.0–94.0	54.32	42.9–65.4	85	66.7
≤4	95.26	91.7–97.6	37.04	26.6–48.5	81.2	73.2
≤5	98.28	95.6–99.5	18.52	10.8–28.7	77.6	78.9
≤6	100	98.4–100	11.11	5.2–20.0	76.3	100
≤7	100	98.4–100	1.23	0.03–6.7	74.4	100
≤8	100	98.4–100	0	0.0–4.5	74.1	100

## Data Availability

The datasets used and/or analyzed during the present study are available from the corresponding author on reasonable request.

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
