# Peer review of "Factors Associated with ICU Admission in Patients with COVID-19: The GOL2DS Score"

_medicina, 2021, doi:10.3390/medicina57121356_

Round 1

Reviewer 1 Report

Thank you for the opportunity to review this manuscript entitled “Factors associated with ICU admission in patients with COVID19: the GOL2DS score”. Authors tried to elaborate on a clinical prognostic score to predict the need for intensive care unit (ICU) admission. However, many unmeasurable variables were not considered. For example, the “admission to ICU” was also influenced by the capacity of ICU bed and availability of stuff, the severity of COVID-19 pandemic in that region, and so on. The major problem of the study is its lack of validation cohort which was important to prove the usefulness of the proposed scores. The generalization of the score was questionable which preclude the application of the score.

Some points may be addressed to improve the manuscript as below.    

  1. As authors stated, pre-existing medical conditions, clinical and laboratory features may predict the disease course. For example, older age, D-dimer concentration higher than 1 μg/mL, and high SOFA scores at the moment of hospitalization are associated with a higher risk of in-hospital death (line 59). However, the manuscript did not even show us the parameters (e.g., SOFA score, d-dimer)
  2. Materials and Methods-> should be placed on next paragraph (page 2, line 71)
  3. Sars-Cov-2à SARS-CoV-2 (page 2, line 75)
  4. Exclusion criteria: patients discharged from or who died in the ER.( This makes the study results questionable, and the study cohort did not include the most severe and most mild patients. Therefore, selection bias exists)
  5. Hospitalized patients “non” requiring intensive care unit during their hospitalization (page 2, line 85)-> change to … “not “requiring…

  1. The important flaws of the statistic : As authors stated, “All variables found statistically significant (p<0.05) at univariate analysis were included in a multivariate logistic regression analysis after correction for confounding factors” (page 3, line 99) à

suggest trying to recalculate and set p<0.1 or <0.2. For example, significant predictors at p-value < 0.2 in a bivariate analysis can be exported to the multivariable logistic regression model. More traditional levels such as 0.05 can fail in identifying variables known to be important (Ref:  Bendel RB, Afifi AA. Comparison of stopping rules in forward regression. Journal of the American Statistical Association. 1977;72:46–53. doi: 10.2307/2286904., Mickey J, Greenland S. A study of the impact of confounder-selection criteria on effect estimation. American Journal of Epidemiology. 1989;129:125–137.)

  1. The most common finding, regarding more than one-half of patients, was interstitial pneumonia. (line 222),blood group A, alveolar oxygenation (FiO2/PaO2), lactate, LDH, dyspnea, SpO2. (line 247)

-> authors may try to analyze the PSI score (pneumonia severity index) for each patients and severity pneumonia index ( e.g., ATS major and minor criteria for severity pneumonia who need to be admitted to ICU) and address the possibility of application. Due to the lack of validation cohort in this study, you may try to validate a pre-existing severity score, which is more feasible for the limited data. (This is just a suggestion)

  1. “We can hypothesize that this group of patients (GOLD2S score >1) can be directly discharged by the ED or admitted to a low-intensity ward”. (line 260)

-> Very questionable because authors had excluded “patients discharged from or who died in the ER”. If possible, you can try including all consecutive patients to avoid the selection bias.

  1. Conclusion paragraph too long

  1. illness despite “di” development of highly effective vaccines (line 269)-> please revise

  1. As authors stated “Among these latter, 5-10% of patients require hospitalization in an ICU” , what is the percentage in the study?

  1. “severe complications like pneumonia, ARDS, and death. (line 278)”

   -> Can authors provide the percentage of above severe complication in the study?

Reviewer 2 Report

Review for manuscript “Factors associated with ICU adission in patients with COVID19: the GOL2DS Score”

This is an interesting study, with good clinical application- namely to identify early patients at risk for a severe course of the disease. The study was performed at the start of the pandemy, in March 2020 and had a good sample size. The methodology is accurate and the performance of the study is very clean, it can easily be understood upon first reading.

Still, there are some issues to adress:

The disease COVID-19 is sometimes written as covid 19, othe times COVID-19 or COVID19 throughout the manuscript- the authors have to keep a single acronym.

The same, the score GOL2DS score is sometimes GOLD2S, GOL2D, GOL2DS throughout the paper- the same, the authors should keep a single acronym. At a certain moment I was wondering if there are two or several scores, but it is only one composite score.

Line 21- “a large proportion of patients require hospitalisation in an ICU”- actually, figures in percentages should be considered “large” – please provide reference or quote.

Line 25- english rephrase- PCR confirmatory for COVID or other way of expressing, PCR should be explained, not acronym on first apperance in text

Introduction is ample, comprehensive and adequate in the same time.

Lines 63-67- is it necessary to focus here on coagulation problems? There are a multitude of complication, including pneumothorax and pneumomediastinum, secondary bacterial infections, etc. why do the authors focus on these, when outcome is actually influenced by many other complications?

Lines 102-103: english rephrase for “was created a ROC curve”

Line 106- better= best? The meaning here should be best

Table 1- explain acronysms in the first line

Line 132- why did the authors chose mean FiO2? Shouldn’t it be more relevand to use median values here? Is the data tested for skewness?

Line 136- please rephrase “did not need for ICU”

Line 137- please delete “for”

Line 144- why did the authors chose to discuss nausea? Because in critically ill patients who are intubated, this symptom cannot be assessed.

Line 169- “…Dyspnea, )”- here the acronym should be decided

Line 173-174- English rephrase- this phrase cannot be understood as it is written

Line 178- “vry”= very

Line 179- “at the contrary”= on the contrary

Table 4- “For each variable have been found”- please rephrase

Punteggio, lattati= please write in English

Figure 5 GOLD2S= GOL2DS? Please decide the acronym of the composite score

The Discussion section does not compare the findings with other already published papers, other predicitive models and methods. The references here are rather scarce and data from the results is rewritten in words. In fact, the Discussion section need much attention to compare the identified results with other similar papers, trace a line for future studies  and use references.

Line 197- how come females are more than males? In line 108- from 313 patients, 103 were female in the results? This has to be clear.

Line 207- please delete “slice”

Line 230- FiO2/pO2- is this the parameter that was used? Not pO2/FiO2??? It can also be found in other places in the manuscript

Line 234- which is the link between high LDH levels and pulmonary involvement?

Lines 240-245- methodology is rephrased here, but should not be

Also 255-259- the same, methodology is rephrased, but should not be present in Discussion

Line 269- di= the

In Conclusions section- lines 266-279 are not the conclusion of this study. The conclusion part should strictly related to the present study findings.

Line 285- “…for validate this score”= “…to validate this score”

Round 2

Reviewer 1 Report

The authors have improved and corrected the manuscript.

Reviewer 2 Report

The authors significantly imporved the manuscript, therefore the paper should be taken into consideration for publication (after minor English language editing)